# Critical role of biomass burning aerosols in enhanced historical Indian Ocean warming

Yiqun Tian[1], Shineng Hu[1] ✉ & Clara Deser [2]

The tropical Indian Ocean (TIO) has experienced enhanced surface warming relative to the tropical mean during the past century, but the underlying mechanisms remain unclear. Here we use single-forcing, large-ensemble coupled model simulations to demonstrate that changes of biomass burning (BMB) aerosols have played a critical role in this TIO relative warming. Although the BMB aerosol changes have little effect on global mean temperatures due to regional cancellation, they significantly influence the pattern of warming over the tropical oceans. The reduction of BMB aerosols over the Indian subcontinent induces a TIO warming, while the increase of BMB aerosols over South America and Africa causes a cooling of the tropical Pacific and Atlantic, respectively. The resultant TIO relative warming leads to prominent global climate changes, including a westward expanded Indo-Pacific warm pool, a fresher TIO due to enhanced rainfall, and an intensified North Atlantic jet stream affecting European hydroclimate.

The tropical oceans have warmed substantially over the past century, with pronounced regional structures (Fig. 1a). Relative to the tropical mean, sea surface temperature (SST) warming is enhanced in the TIO, western Pacific and eastern Atlantic, and suppressed in the central-eastern Pacific. The enhanced warming in the climatologically warm TIO has attracted much attention[1–3]. as it has been suggested to cause a wide range of climate impacts via teleconnections, including a strengthened positive phase of North Atlantic Oscillation (NAO)[4], reduced Sahel rainfall[5], an enhanced Pacific Walker Circulation[6], the occurrence of a North Atlantic "warming hole"[7], and an intensified Atlantic Meridional Overturning Circulation[8].

The enhanced TIO warming relative to the tropical mean is a robust feature across different datasets (Supplementary Figs. 1 and 2) and is particularly evident after the 1950s (Supplementary Fig. 3). We find that the observed absolute TIO warming rate is subject to large data uncertainties, but is positively correlated with the tropical mean warming rate (Fig. 1c). One common feature to all the observational datasets investigated here is that the TIO warming is greater than the tropical-mean warming (Fig. 1d). Our analysis suggests that this TIO "relative warming" trend remains a prominent feature regardless of the time period chosen for analysis (Supplementary Fig. 2). The fact that the TIO has warmed faster than the tropical oceans on average may imply an overall increase of TIO rainfall, therefore driving the TIO-induced global teleconnections mentioned above, but whether TIO rainfall has indeed increased over the past century remains under debate due partly to the lack of direct observations[9,10].

The physical mechanisms underlying the observed TIO relative warming remain unclear. Several mechanisms have been proposed to explain the TIO absolute warming, including, for example, the increase in greenhouse gas concentration[1,3] or changes in El Niño-Southern Oscillation properties[2]. These mechanisms may not necessarily be applicable to TIO relative warming, however. Given the critical role of relative SST changes in shaping the tropical rainfall pattern[11], it is therefore important to address the following questions. Is the observed TIO relative warming a response to external radiative forcing? Does internal climate variability play a role? If externally forced, which radiative forcing agent (or agents) is responsible for the TIO relative warming? These questions are difficult to tackle by relying solely on observations. To that end, we analyze single-forcing, large-ensemble coupled model simulations together with available observations, which has led us to identify a previously overlooked role for BMB aerosol forcing in the historical TIO relative warming.

[1]Division of Earth and Climate Sciences, Nicholas School of the Environment, Duke University, Durham, NC, USA. [2]Climate and Global Dynamics, National Center for Atmospheric Research, Boulder, CO, USA. ✉e-mail: shineng.hu@duke.edu

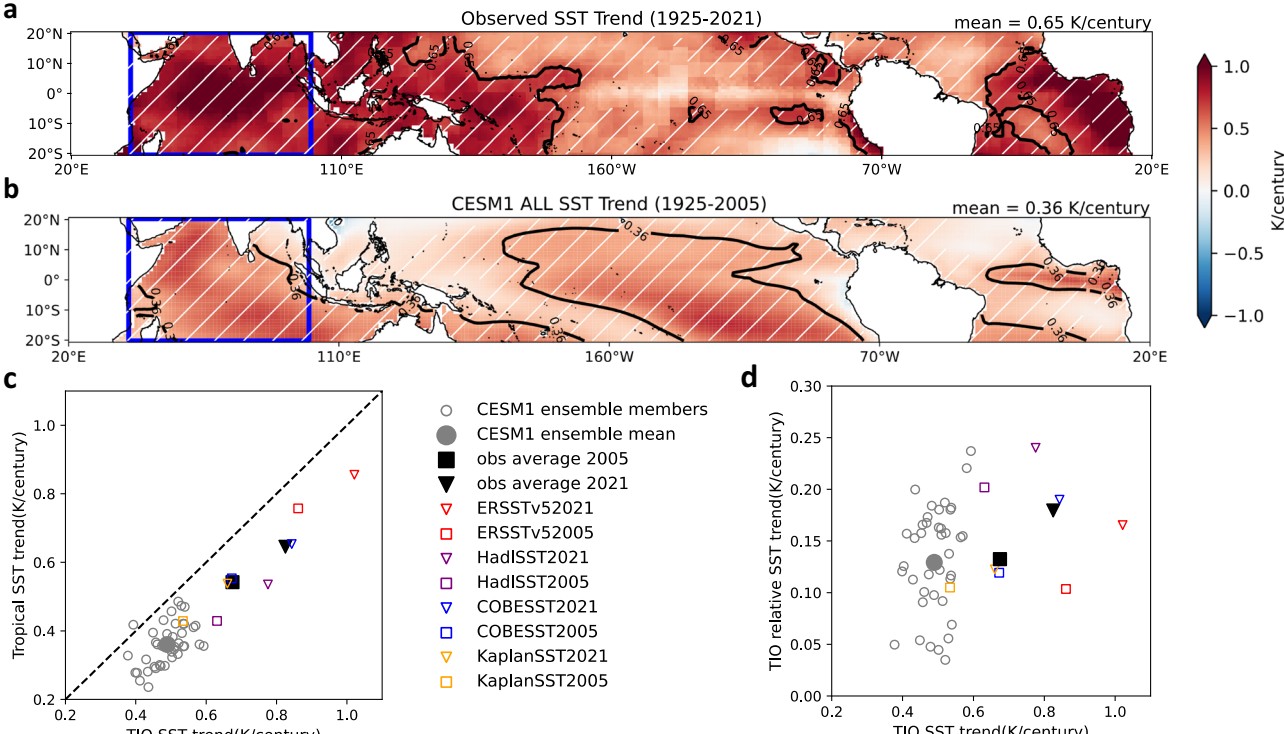

**Fig. 1 | Observed and model simulated tropical sea surface temperature (SST) trends since 1925. a** Observed tropical SST trends (K/century) during 1925–2021, averaged across four SST datasets (ERSSTv5, HadISST, COBE-SST, Kaplan SST). **b** Community Earth System Model v1 (CESM1)-Large Ensemble (LE) ensemble-mean tropical SST trends (K/century) during 1925–2005 from the all-forcing experiments. In panels a and b, the black contour line represents the isopleth of the tropical-mean SST trend (values shown on the top-right corner). **c** Simulated and observed Tropical Indian Ocean (TIO) SST trends versus tropical-mean SST trends during 1925–2005. The observed trends during 1925–2021 are also shown for comparison. **d** Similar to **c** but for TIO SST trends vs. TIO relative SST trends. Relative SST is defined as absolute SST minus tropical-mean SST. White hatches in **a** and **b** represent the regions that are 99% significant based on a *t*-test.

## Results

### Tropical SST response to historical BMB aerosol changes

To explore the mechanisms of TIO warming, we use the set of initial-condition Large Ensembles (LEs) conducted with Community Earth System Model version 1 (CESM1) (Methods). CESM1 is a widely used, fully coupled general circulation model (GCM) that participates in the Coupled Model Intercomparison Project Phase 5 (CMIP5). The LE setting helps to separate the externally forced climate response from internal climate variability[12]. In addition to the standard all-forcing (ALL) historical LE simulations[13], we use the "all-but-one-forcing" LE simulations[14] to isolate the impacts of greenhouse gases (GHG), anthropogenic aerosols (AAER), and BMB aerosols. The historical portion of the simulations spans the period 1920–2005, and we use the period 1925–2005 for analysis to remove some of the potential influence of ocean initial condition memory. The main conclusions presented do not change if a different period (e.g., 1930–2005) is used for analysis.

A recent study suggests that CMIP5 coupled climate models on average simulate largely uniform surface warming over the tropical oceans and thus underestimate the historical TIO relative warming rate compared to observations[15]. Although the ensemble-mean of the CESM1 ALL LE underestimates the observed tropical-mean ocean warming rate (Fig. 1c), it captures the TIO relative warming with good fidelity (Fig. 1b). This implies that the observed TIO relative warming may be partly attributable to external forcing, according to CESM1. The observation-CESM1 mismatches (e.g., the enhanced western Pacific warming) can result from multidecadal internal variability, among other factors such as uncertainties in radiative forcing and model mean-state biases. We note that the LE's ensemble spread in simulated TIO SST variations, both absolute and relative, encompasses the observed variations (Supplementary Fig. 3). Although the LE ensemble

seems to slightly underestimate the TIO absolute warming since 1925, its spread well encompasses the observed TIO relative warming (Fig. 1d). These results indicate that the model has a realistic depiction of the combined influences of natural variability and forced response in TIO trends. Interestingly, we find that ensemble members that simulate stronger TIO relative warming rates tend to show better agreement with the observed pattern of tropical warming, quantified in terms of spatial correlation (Supplementary Fig. 4).

Given the complex set of radiative forcing changes over the past century, is there a primary factor that drives the forced component of TIO relative warming in CESM1? To answer this question, we investigate the impacts of individual forcing agents, including GHG, AAER, and BMB aerosols using the "all-but-one-forcing" LEs. As expected, GHG and AAER are the two dominant and partially offsetting drivers of tropical mean surface temperature trends during the past century, while BMB aerosols play a minor role (Fig. 2a). Next, we compute the standard deviation of all the grid points in the ensemble-mean SST trend field and use it as a measure of spatial variance. Interestingly, we find that BMB aerosols contribute significantly to the spatial variability of tropical SST warming and is comparable to the contributions from GHG or AAER (Fig. 2b). As far as we know, the impact of BMB aerosols on the spatial variability of tropical SST trends (i.e., the pattern) has been overlooked in previous studies. More specifically, we find that BMB aerosol changes serve as the main driver of the TIO relative warming in the historical period, as we elaborate below.

Figure 3b–f shows the ensemble-mean 1925–2005 trends of tropical relative SST (i.e., absolute SST minus tropical-mean SST; see Methods), associated with each specific radiative forcing. The standard, all-forcing historical simulations exhibit a complex spatial structure with relative warming located in the central equatorial

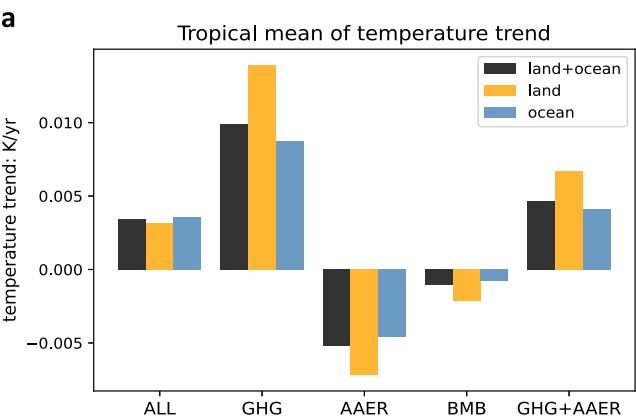

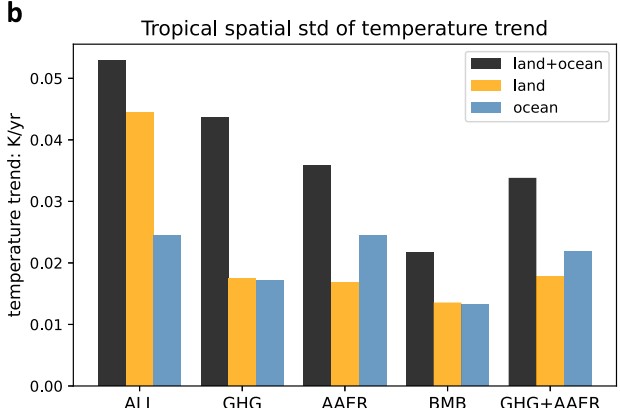

**Fig. 2 | Attribution of tropical warming to individual radiative forcing agents in 1925–2005. a** Ensemble-mean, tropical-mean surface temperature trends (K/year) attributed to each radiative forcing: all-forcing (ALL), greenhouse gases (GHG), anthropogenic aerosols (AAER), BMB aerosols, anthropogenic aerosols and greenhouse gas combined (AAER + GHG). For each forcing, calculations have been performed for land only, ocean only, and land-ocean combined. **b** Similar to panel a but for spatial standard deviation of ensemble-mean SST trends in the tropics. For both panels, the tropics is defined as 20°S–20°N.

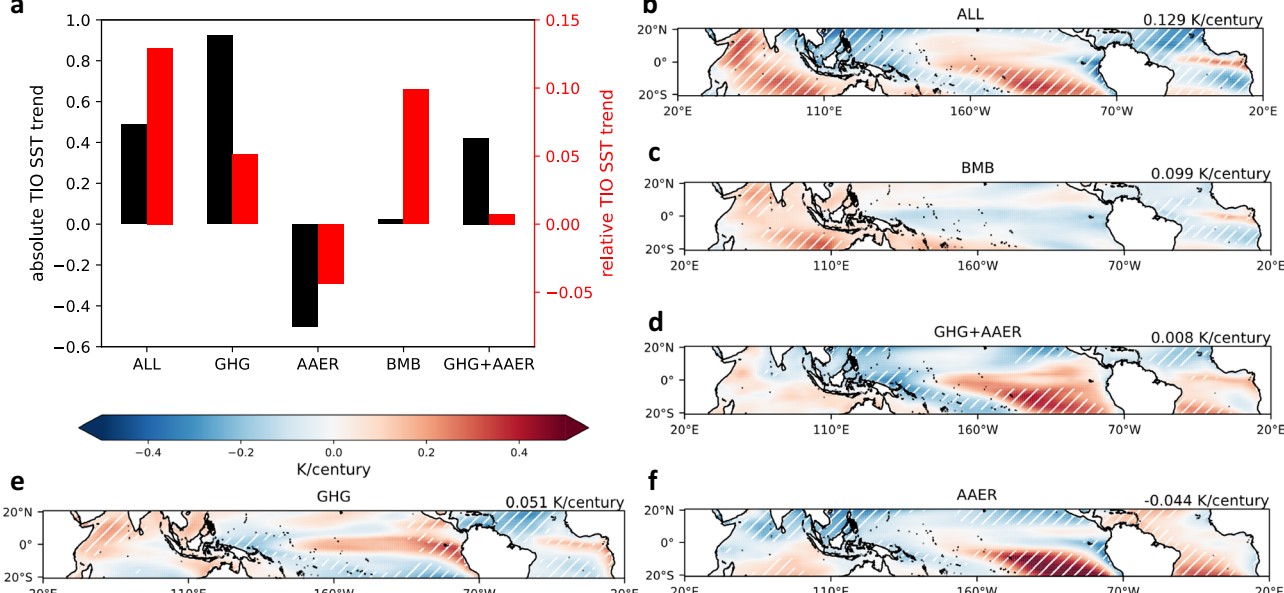

**Fig. 3 | Critical role of biomass burning (BMB) aerosols in the historical tropical Indian Ocean (TIO) relative warming. a** Absolute and relative TIO sea surface temperature (SST) trends (K/century) during 1925–2005 attributed to individual radiative forcing agents, similar to Fig. 2. Note the two different vertical axes. **b–f** Tropical relative SST trends (K/century) during 1925–2005 in all-forcing experiments and attributed to each radiative forcing. Relative SST is defined as absolute SST minus tropical-mean SST. The value on the top right corner represents the TIO-mean relative SST trend. White hatches in **b–f** represent the regions that are 99% significant based on a *t*-test.

Pacific, the southeastern tropical Pacific, the central-eastern equatorial Atlantic, and almost the entire TIO (Fig. 3b). Our forcing decomposition analysis suggests that although BMB aerosols contribute little to the absolute TIO SST trend, it acts as the most important radiative forcing for the TIO relative warming (Fig. 3a). In contrast, GHG and AAER are two leading factors for TIO absolute SST changes, consistent with previous studies[3], but not for TIO relative SST. Within the TIO, both GHG and AAER induce a hemispheric asymmetry in relative SST trends, but these patterns largely cancel out leaving only weak net changes (Fig. 3d–f). In the Pacific and Atlantic, GHG leads to an equatorially enhanced warming, a common feature identified in coupled GCMs[16,17]. The southeastern tropical Pacific relative warming is mainly induced by AAER, which is not unexpected given the greater anthropogenic aerosol loading in the Northern Hemisphere[18,19]. Below we will focus primarily on the mechanisms and impacts of BMB aerosol-induced TIO relative warming.

**Mechanisms of BMB aerosol-induced TIO relative warming**

How do the historical BMB changes affect the tropical SST warming pattern? BMB aerosols alter the radiative budget both at the top-of-atmosphere and the surface by absorbing and scattering solar radiation directly and by influencing cloud properties via indirect and semi-direct effects[20]. The net radiative forcing of BMB aerosols at the ocean surface is suggested to be negative, albeit bearing large uncertainties, and it acts to cool the ocean. During 1925–2005, BMB aerosols have significantly increased over Equatorial Asia, South America and Central Africa, but have greatly decreased over the Indian subcontinent, primarily in the 1950s (Supplementary Fig. 5b)[21]. These BMB aerosol changes which originate over land in turn affect the surrounding tropical oceans, leading to an increase of aerosol optical depth (AOD) over the tropical Atlantic, the tropical Pacific, and the far eastern TIO, and a decrease of AOD over most of the TIO (Supplementary Fig. 5a). This spatial heterogeneity in BMB emissions gives rise to a pronounced structure in the

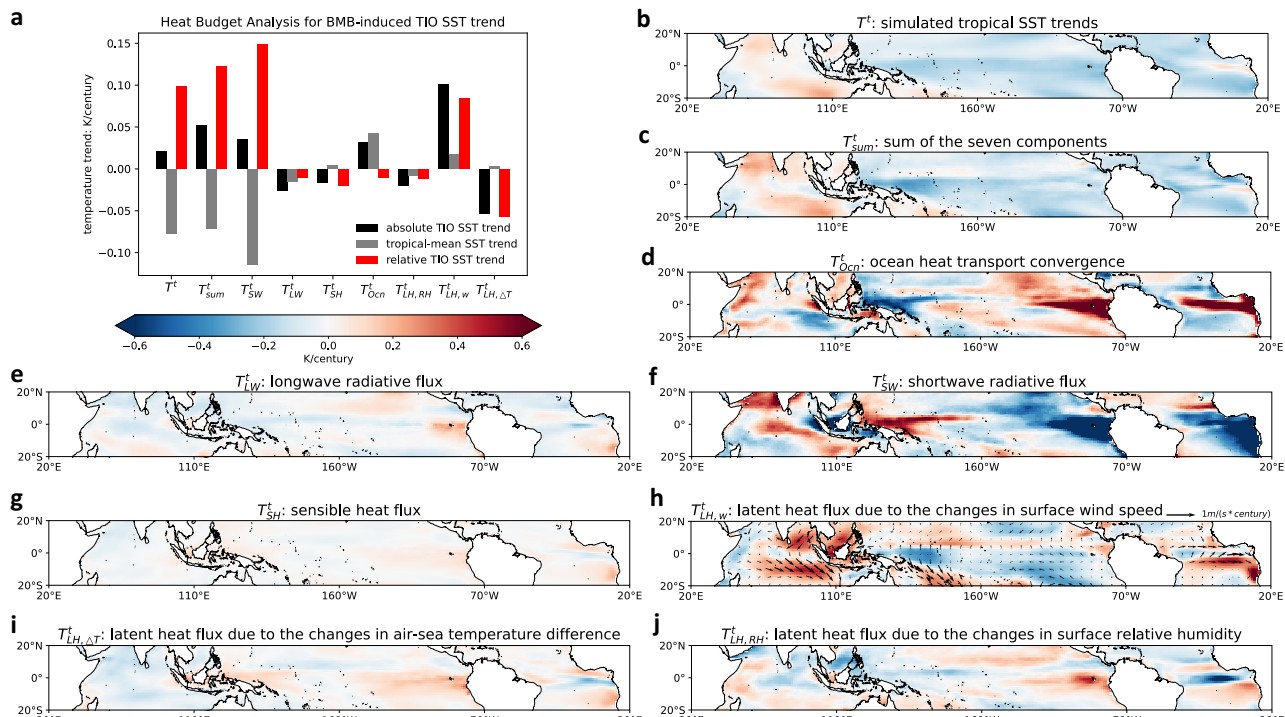

**Fig. 4 | Heat budget analysis for biomass burning (BMB) aerosol-induced Tropical Indian Ocean (TIO) warming. a** TIO sea surface temperature (SST) trends (K/century) during 1925–2005 attributed to each component in the ocean-mixed layer heat budget (Methods). The TIO absolute trends are further split into two components, tropical-mean SST trends and TIO relative SST trends. **b** Simulated tropical SST trends during 1925–2005 (K/century) induced by BMB changes. **c–j** BMB aerosol-induced tropical SST trends during 1925–2005 attributed to **c** sum of the following seven components ($T_{sum}^t$), the changes in **d** ocean heat transport convergence ($T_{Ocn}^t$), **e** longwave radiative flux ($T_{LW}^t$), **f** shortwave radiative flux ($T_{SW}^t$), **g** sensible heat flux ($T_{SH}^t$), and latent heat flux due to the changes in **h** surface wind speed ($T_{LH,w}^t$), **i** air-sea temperature difference ($T_{LH,\Delta T}^t$), and **j** surface relative humidity ($T_{LH,RH}^t$).

SST response, with relative warming over the TIO and relative cooling over the tropical Pacific and Atlantic (Fig. 4b).

To uncover the atmospheric and oceanic adjustments to the BMB aerosol forcing, we conduct an ocean mixed layer heat budget analysis following ref. 22. (Methods). Below we summarize the key points of the methodology. The SST changes induced by external radiative forcing (here BMB) can be decomposed as:

$$T^t \approx T_{SW}^t + T_{LW}^t + T_{SH}^t + T_{Ocn}^t + T_{LH,w}^t + T_{LH,RH}^t + T_{LH,\Delta T}^t \quad (1)$$

In Eq. (1), the simulated, ensemble-mean SST trend, denoted as $T^t$, is broken down approximately into 7 components. It is worth noting that the sum of the 7 sub-terms (denoted as $T_{sum}^t$) may not be expected to exactly match the model simulated $T^t$ due to the assumptions made through the derivation, e.g., a linearized bulk formula of latent heat flux (Methods). The first 4 components represent the contributions from the long-term trends in shortwave radiative flux $T_{SW}^t$, longwave radiative flux $T_{LW}^t$, sensible heat flux $T_{SH}^t$, and ocean dynamics $T_{Ocn}^t$, respectively. The last three components represent the contributions from the trends in surface wind speed $T_{LH,w}^t$, relative humidity $T_{LH,RH}^t$, and air-sea temperature gradient $T_{LH,\Delta T}^t$, respectively, via latent heat flux changes. The full derivations and the expression of each component are shown in the Methods.

The sum of the 7 SST trend components ($T_{sum}^t$) agrees quite well with the tropical SST trend response to BMB forcing (i.e., $T^t$) with a pattern correlation of 0.88, validating our methodology (Fig. 4b, c). The biggest contributors to the BMB-induced TIO relative warming are $T_{SW}^t$ and $T_{LH,w}^t$, while $T_{Ocn}^t$ contributes to the TIO absolute but not relative warming (Fig. 4a), as we will further discuss in detail below.

- $T_{SW}^t$: To separate the BMB aerosol's direct and indirect/semi-direct effects, we split $T_{SW}^t$ into the clear-sky and cloud components (Fig. 4f; Supplementary Fig. 6). The BMB reduction over the TIO increases the clear-sky shortwave radiation reaching the surface and induces a local warming especially around the Indian subcontinent (see Supplementary Fig. 5), while cloud changes contribute to the warming primarily near the equator. At the same time, the BMB aerosol increase over the eastern Pacific and Atlantic cools these areas primarily via clear-sky shortwave radiation, amplified by SW cloud feedbacks in the eastern Pacific. Altogether, $T_{SW}^t$ becomes the primary driver of the BMB aerosol-induced TIO relative warming (Fig. 4a).

- $T_{LH,w}^t$: The anomalous easterly winds in the northern TIO and northwesterly winds in the southern TIO oppose the background wind and cause decreases in surface wind speed and thus latent heat flux, acting as a warming effect (Fig. 4h). In addition, a considerable cooling due to the increase of wind speed is identified over the southeast subtropical Pacific. In terms of TIO relative warming, $T_{LH,w}^t$ serves as the second largest contributor (Fig. 4a).

- $T_{Ocn}^t$: The change in ocean heat transport induces a relatively complex warming/cooling structure (Fig. 4d). The prominent warming in the western TIO is consistent with the anomalous alongshore northerly winds that are expected to induce Ekman convergence and downwelling near the coast. The equatorial Atlantic warming is associated with an anomalous Ekman convergence due to the westerly wind anomalies, while the eastern equatorial Pacific warming can potentially be explained by a weaker poleward heat transport by the climatological Ekman divergence due to the cooling SSTs (Fig. 4b). As a result, ocean heat transport changes only play a minor role in the TIO relative SST changes (Fig. 4a).

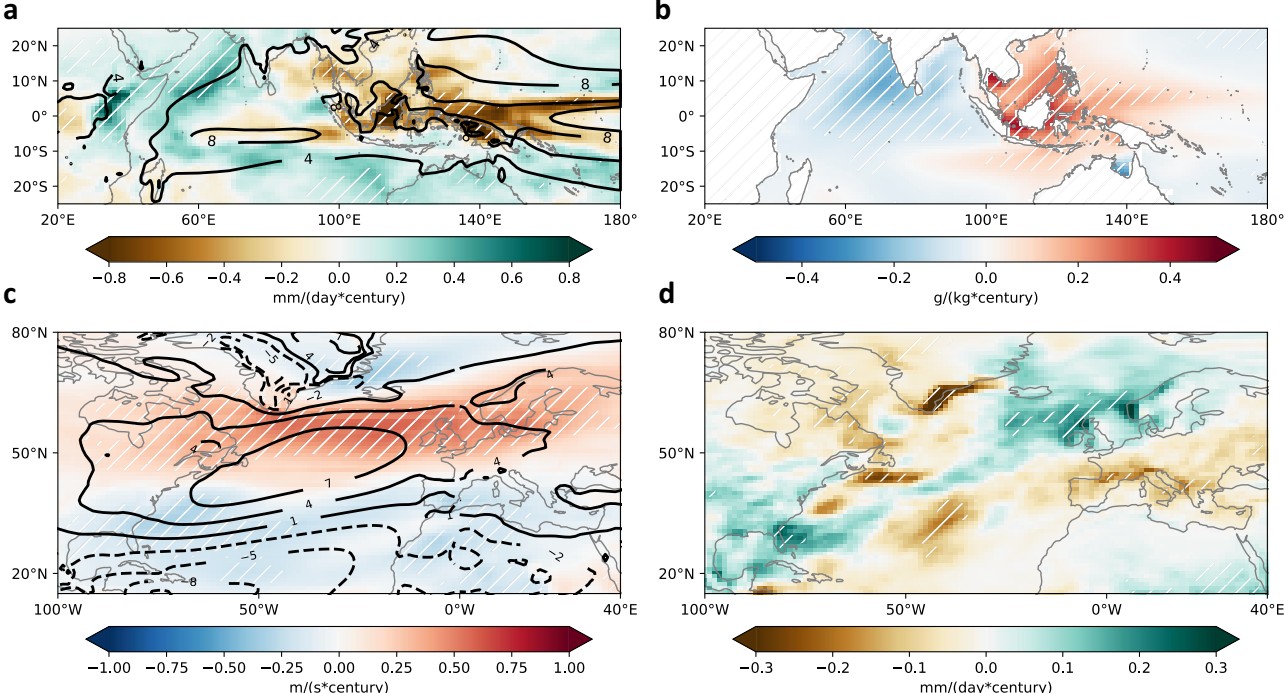

**Fig. 5 | Potential climate impacts associated with biomass burning (BMB) aerosol-induced Tropical Indian Ocean (TIO) relative warming. a** BMB aerosol-induced rainfall trends (colors; mm/d/century) overlaid by climatological rainfall (contours; mm/d) and **b** BMB aerosol-induced ocean salinity trends (g/kg/century) during 1925–2005. **c** BMB aerosol-induced 850 mb zonal wind trends (colors; m/s/century) overlaid by climatological 850 mb zonal wind (contours; m/s) and **d** BMB aerosol-induced rainfall trends (mm/d/century) during 1925–2005. White hatches in **a**–**d** represent the regions that are 99% significant based on a *t*-test.

The changes in air-sea temperature gradient generally act as a damping effect to the tropical SST changes with a pattern correlation of −0.67 (Fig. 4i; see Fig. 4b), since a warmer SST tends to enhance the air-sea temperature contrast favoring an enhanced latent heat release. Other terms in the heat budget are relatively small in terms of TIO basin averaged SST changes, although they can be non-negligible locally.

**Potential impacts of BMB aerosol-induced TIO relative warming**

The BMB aerosol-induced TIO relative warming can potentially have important impacts on regional and global climate. First, the TIO relative warming leads to an increase of rainfall over most of the TIO, while in contrast, the western Pacific cooling in response to BMB aerosol increases over Equatorial Asia induces a local reduction in precipitation (Fig. 5a). Such a zonal dipole in rainfall anomalies suggests a westward displacement of the Indo-western Pacific warm pool convective center, although the magnitude of rainfall change is rather weak (~10% of the climatological rate over a century). Consistent with the precipitation change dipole, the TIO becomes fresher while the western Pacific becomes saltier (Fig. 5b). These changes are also seen in the all-forcing experiments but with a greater amplitude (Supplementary Fig. 7), implying that BMB changes may play a critical role in shaping the Indo-western Pacific warm pool rainfall change pattern in the past century.

The BMB aerosol-induced local rainfall changes over the TIO and western Pacific can potentially lead to far-reaching impacts via global teleconnections. For example, we find that the North Atlantic jet stream is significantly enhanced and shifted northward by the BMB aerosol forcing (Fig. 5c), similar to the jet strengthening seen in the all-forcing experiments (Supplementary Fig. 7). The enhanced North Atlantic jet stream transports more water vapor towards northwestern Europe, enhancing rainfall over the British Isles and Scandinavia and reducing it over southern Europe and the Mediterranean (Fig. 5d; Supplementary Fig. 7). The BMB aerosol-induced changes in the North Atlantic jet and the associated European rainfall pattern all

suggest a shift towards the positive phase of NAO, as observed in the past century[23]. The BMB aerosol-induced changes can explain about a half of the NAO positive trend seen in the all-forcing experiments, while the other half comes mainly from GHG-induced changes (Supplementary Fig. 8). Our results are broadly consistent with the previous studies that argue a TIO warming can induce a positive NAO[4,7,8]. A more definitive demonstration of the BMB aerosol-TIO-NAO linkage, particularly whether the BMB aerosol-induced Atlantic jet strengthening is explained solely by the TIO relative warming or partially by other factors, requires further investigation.

## Discussion

Our findings with CESM1 "all-but-one-forcing" large ensembles imply that BMB aerosols may play a more important role in historical climate change than previously thought. Although the BMB aerosol impact on global mean temperature change may be secondary as compared with GHG or AAER, it does significantly influence the pattern of warming. In fact, over 80% of global BMB aerosol emissions have occurred in tropical regions[24], and we indeed find that the change in BMB aerosol is critical in shaping the tropical SST warming pattern in the past century. In particular, the BMB aerosol change is the dominant contributor to the forced component of TIO relative warming, at least compared to GHG and AAER (Fig. 3; Supplementary Fig. 9). While observational data products also show relative TIO warming, most of this warming occurs after 1960 with considerable uncertainty in terms of magnitude (Supplementary Fig. 9). It is likely that internal variability also contributes to the observed relative TIO SST record. The TIO relative warming induced by the BMB aerosol changes leads to locally enhanced rainfall, which together with the BMB aerosol-induced cooling of the western Pacific, potentially acts to shift the Indo-western Pacific warm pool convective center westward. The BMB aerosol-induced tropical rainfall changes may in turn cause a positive NAO phase and influence North Atlantic/European climate via atmospheric teleconnections. To what extent the results presented in this study are

model dependent needs to be tested by other climate models conducting similar types of historical experiments isolating the contribution of BMB aerosol changes, which is currently not a protocol for most models.

Our study points to an urgent need to accurately represent the chemical, microphysical, and radiative properties of BMB aerosols in GCMs. In the past decades, enormous progress has been made in BMB-related studies, including both field campaign observations[25–30] and the associated modeling efforts[31–33]. Despite these tremendous advances, an accurate representation of the influence of BMB aerosols in climate models remains challenging. For example, a recent study suggests that BMB aerosols in most climate models are too absorbing, leading to a potential bias in radiative forcing[34]. In contrast, another study argues that the absorbing nature of BMB aerosols over the Southeast Atlantic is underestimated in most climate models[35]. Such discrepancies have yet to be reconciled. Another recent study reports that a discontinuity in the variability of prescribed BMB aerosol emissions between the satellite era and the prior period in some climate models can rectify the modeled mean climate and generate a spurious warming[36]. As continuous observational efforts remain necessary in the coming decades, developing proxies of historical BMB aerosols[37] and modeling their physical and chemical properties, transport within the atmosphere, interactions with clouds, and role in carbon cycle dynamics all need to be refined. In that regard, modeling efforts like the Fire Modeling Intercomparison Project[33] will be particularly helpful in interpreting and potentially reducing inter-model spread. Improving the understanding and simulation of BMB aerosols thus requires the integrated efforts of worldwide observational and laboratory communities, Earth science modeling communities, and more importantly, the collaboration between those communities.

## Methods

### Observational datasets

We use four monthly observational SST datasets: (1) National Oceanic and Atmospheric Administration Extended Reconstructed Sea Surface Temperature Version 5 (ERSSTv5) with resolution $2° \times 2°$[38]. (2) Hadley Centre Sea Ice and SST v.1.1 (HadISST 1.1) with resolution $1° \times 1°$[39]. (3) Centennial In Situ Observation-Based Estimates of the Variability of SST and Marine Meteorological Variables (COBE) with resolution $1° \times 1°$[40]. (4) Kaplan Extended SST v2 with resolution $5° \times 5°$[41].

### CESM1 large-ensemble simulations

We use four sets of simulations from the Community Earth System Model version 1 Large Ensemble (CESM1-LE)[13] produced by the National Center for Atmospheric Research (NCAR). The first set is the standard all-forcing (ALL) historical simulations which contains all historical radiative forcings; it has 40 ensemble members. The other three sets are the CESM1-LE "all-but-one-forcing" experiments[14], which are designed to isolate the influence of individual forcing agents, including anthropogenic aerosols (AAER), greenhouse gases (GHG), and biomass burning aerosols (BMB). They are identical to the all-forcing historical simulations (i.e., ALL) except that one forcing agent is fixed at its 1920 level. More specifically, the XGHG (where "X" refers to the fact that GHG is held fixed) set has 20 ensemble members with fixed GHG; the XAAER set has 20 ensemble members with fixed AAER; the XBMB set has 15 ensemble members with fixed BMB aerosols. Each ensemble member within each set starts from slightly different atmospheric initial conditions on the order of $10^{-14}$K. The influence of each individual forcing agent can therefore be determined by subtracting "all-but-one-forcing" member-mean from ALL member-mean, that is,

GHG = ALL ensemble mean − XGHG ensemble mean
AAER = ALL ensemble mean − XAAER ensemble mean
BMB = ALL ensemble mean − XBMB ensemble mean

The full time period for all the experiments under historical forcing is 1920–2005. To avoid the potential influence of ocean initial conditions, we use the period 1925–2005.

### Definitions of indices

For both observations and simulations, the tropics refers to the latitudinal band 20°S–20°N, and the TIO refers to the region 20°S–20°N, 40°E–100°E. Relative SST is defined as the difference between local SST and tropical mean SST.

The NAO pattern and index used here are computed as the first EOF and PC of annual mean sea level pressure anomalies within the region of 20°N-80°N, 90°W-40°E, after concatenating all the members from ALL, XBMB, XAER and XGHG.

### Ocean mixed layer heat budget analysis

To further understand the mechanisms of BMB-induced TIO warming, we conducted ocean mixed layer heat budget analysis following the methodology in ref. [22]. and the references therein. The ocean surface mixed layer heat budget can be written as:

$$\rho C_p H \frac{\partial SST}{\partial t} = SW + LW + SH + LH + Ocn. \tag{2}$$

The left-hand side represents mixed layer heat storage, where $\rho$ is density of sea water, $C_p$ is specific heat of sea water, H is depth of ocean mixed layer and SST is temperature of ocean mixed layer. On the right-hand side, SW is surface net shortwave radiative flux, LW is surface net longwave flux, SH is surface sensible heat flux, LH is surface latent heat flux and Ocn is the heat flux due to ocean dynamics. We then do the linear regression onto time to the variables in Eq. (2). Since the trend of ocean mixed layer heat storage can be treated as negligible[11,42], the left-hand side becomes zero and we have:

$$0 = SW^t + LW^t + SH^t + LH^t + Ocn^t \tag{3}$$

We define downward as positive and the superscript t represents BMB induced ensemble mean trend during 1925–2005. Latent heat flux term is directly related to SST and can be written as:

$$LH = -\rho_{air} L_v C_E W \left[ q^*(SST) - q_{air} \right] \tag{4}$$

$q_{air}$ in Eq. (4) is specific humidity of air above sea surface. It can be expressed as:

$$q_{air} = RH q^*(SST - \Delta T) \tag{5}$$

Where RH is relative humidity at sea surface, $\Delta T$ is temperature gradient near sea surface and we define it as $(SST - T_{air})$. We further rewrite $q_{air}$ by using Clausius-Clapeyron equation:

$$q_{air} = RH q^*(SST) e^{-\alpha \Delta T} \tag{6}$$

Where $\alpha \approx 0.06/K$. We plug Eq. (6) into Eq. (4) and get new latent heat flux expression:

$$LH = -\rho_{air} L_v C_E W q^*(SST)(1 - RH e^{-\alpha \Delta T}) \tag{7}$$

Next, we use linear regression to get $LH^t$ from Eq. (7):

$$
\begin{aligned}
LH^t &= \frac{\partial LH}{\partial W} W^t + \frac{\partial LH}{\partial SST} SST^t + \frac{\partial LH}{\partial RH} RH^t + \frac{\partial LH}{\partial \Delta T} \Delta T^t \\
&= \frac{\overline{LH}}{\overline{W}} W^t + \alpha \overline{LH} SST^t - \frac{\overline{LH}}{e^{\alpha \overline{\Delta T}} - \overline{RH}} RH^t + \frac{\alpha \overline{LH}\, \overline{RH}}{e^{\alpha \overline{\Delta T}} - \overline{RH}} \Delta T^t
\end{aligned}
\tag{8}
$$

where the overbar for $\overline{LH}, \overline{W}, \overline{\Delta T}, \overline{RH}$ is climatology computed using the period of 1925–1944. Plugging in Eq. (8), we can then rewrite Eq. (3) as:

$$T^t \approx -\frac{SW^t + LW^t + SH^t + Ocn^t}{\alpha \overline{LH}} - \frac{W^t}{\alpha \overline{W}} - \frac{\overline{RH} \Delta T^t}{e^{\alpha \overline{\Delta T}} - \overline{RH}} + \frac{RH^t}{\alpha(e^{\alpha \overline{\Delta T}} - \overline{RH})}$$

$$= T^t_{SW} + T^t_{LW} + T^t_{SH} + T^t_{Ocn} + T^t_{LH,w} + T^t_{LH,RH} + T^t_{LH,\Delta T}$$

(9)

where $SST^t$ is written as $T^t$ for brevity. The last three terms are all related to latent heat flux changes via the changes in surface wind speed $T^t_{LH,w}$, relative humidity $T^t_{LH,RH}$ and temperature gradient $T^t_{LH,\Delta T}$, respectively. In summary, $T^t$ can be split into seven sub-terms as follow:

$$T^t_{SW} = -\frac{SW^t}{\alpha \overline{LH}}$$

(10)

$$T^t_{LW} = -\frac{LW^t}{\alpha \overline{LH}}$$

(11)

$$T^t_{SH} = -\frac{SH^t}{\alpha \overline{LH}}$$

(12)

$$T^t_{Ocn} = -\frac{Ocn^t}{\alpha \overline{LH}}$$

(13)

$$T^t_{LH,w} = -\frac{W^t}{\alpha \overline{W}}$$

(14)

$$T^t_{LH,\Delta T} = -\frac{\overline{RH} \Delta T^t}{e^{\alpha \overline{\Delta T}} - \overline{RH}}$$

(15)

$$T^t_{LH,RH} = \frac{RH^t}{\alpha(e^{\alpha \overline{\Delta T}} - \overline{RH})}$$

(16)

It is worth noting that the sum of the seven sub-terms ($T^t_{sum}$) may not be expected to exactly match the model simulated $T^t$ due to the assumptions made through the derivation, e.g., a linearized bulk formula of latent heat flux.

## Data availability

All data used in this study are publicly available. For observational datasets, the NOAA's ERSSTv5 data are available at https://psl.noaa.gov/data/gridded/data.noaa.ersst.v5.html; HadISST 1.1 data at https://www.metoffice.gov.uk/hadobs/hadisst/data/download.html; COBE SST at https://psl.noaa.gov/data/gridded/data.cobe.html; Kaplan Extended SST v2 at https://psl.noaa.gov/data/gridded/data.kaplan_sst.html. CESM1-LE data are available through the National Center for Atmospheric Research (NCAR) Casper cluster at /glade/campaign/cesm/collections/cesmLE/CESM-CAM5-BGC-LE/ or through web access by guidance: https://www.cesm.ucar.edu/working-groups/climate/simulations/cesm1-single-forcing-le.

## Code availability

All codes are available upon request.

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

## Acknowledgements

S.H. is supported by NASA Award 80NSSC22K1025. We acknowledge computational support from the NCAR-Wyoming Supercomputing Center. We acknowledge Isla Simpson for the help with the analysis of aerosol forcing data for CESM1 LE. NCAR is sponsored by the National Science Foundation.

## Author contributions

Y.T. and S.H. conceived the original idea and wrote the first draft of the manuscript. Y.T. performed the analysis. Y.T., S.H., and C.D. contributed to the interpretation of results and edited the manuscript.

## Competing interests

The authors declare no competing interests.
