## [Peer Review File · Nature Communications]

Critical role of biomass burning aerosols in enhanced historical Indian Ocean warmingREVIEWER COMMENTS

Reviewer #1 (Remarks to the Author):

Present study investigates the possible reasons for the relative stronger surface warming over the tropical Indian Ocean than the tropical mean during the past century, and finds the important contribution from the changes of biomass burning (BMB) aerosols. The results are really interesting, and emphasizing the role of BMB is quite new and important to the climatology field. The whole paper is well organized, but I still have some concerns before the publication of this paper.

Major comments:

1. My major concern is the model, and only one model is used in present study. The model may have some problems in simulating the physical processes involved even if CESM1-LE can capture some of observational warming trend, which could be only a result of model bias compensation. Besides, as the authors mentioned, the aerosol's direct and indirect effects are not well simulated, and aerosol is the major source of forcing uncertainty in IPCC report. Thus, whether the obtained results are model-dependent? Does there exist any other experiments or models that could be used to further repeat your results (i.e., PDRMIP)? To what extent the BMB contributes to the TIO relative warming could be credible?

2. Why is the role of BMB critical in the TIO relative warming? As shown in Fig. 3, the total aerosols' contribution is negative, even if the large positive contribution shown by the BMB. That is to say, the other aerosols, i.e., black carbon, should be the larger negative contribution than the positive contribution of BMB. Besides, why is the combined trend of GHG and aerosols far less than the original TIO relative warming trend? There seems to exist other reasons, which work and make up the large gap between the combined trend of GHG and aerosols and original TIO relative warming trend.

Minor comments:

L. 88-89: Remove "the relative warming of".

L. 89-90: Why is the relative warming over the tropical western Pacific not well captured? Please explain and discuss.

L. 265: "greenhouse gases, (GHG)" should be "greenhouse gases (GHG),"

Reviewer #2 (Remarks to the Author):

I reviewed the manuscript with a focus on the attribution to the TIO relative warming due to changes in regional BMB aerosol emissions. The physical mechanisms for TIO warming due to regional changes in BMB aerosol emissions are well founded, and the authors acknowledge some of the ongoing areas of study related to BMB absorption. Studying the effect of changing BMB aerosol emissions on the temperature trends of TIO relative to the tropical oceans in general is a useful method. In general, I find the authors largely remained focused on exploring a singular question, with relatively straight-forward but interesting methods and findings, and solid visualizations/figures of the key elements contributing to the overall conclusions. My recommendation is that it should be published after minor revisions.

Simple comments:

Line 61 "rise of greenhouse gases" \diamond "increase in greenhouse gas concentration"; rise makes it sound like the greenhouse gases molecules are floating upwards.

Line 136 "reduced" \diamond "decreased"

Lines 159-160: Would it be clearer to state that this analysis here is for BMB-induced TIO relative warming as opposed to just TIO relative warming?

Figure 4: In b-j, I suggest writing out the terms in the subtitles of the figures rather than just using the variable as the subtitle. For example, in Figure 4b, the subtitle would be "Simulated tropical SST trends induced by BMB changes (T_t)". You have plenty of room to include this and that way the readers would largely be able to learn from the figure itself rather than having to connect the individual variables to the caption.

References seem misaligned. Line 212 references ref24 when I think the authors mean ref25. Similarly, in the Discussion, the 80% of BB emissions statement in line 222 likely refers to Van Marle et al 2017 (ref26) but the authors cite ref25. Seems like everything is off by one reference number for some reason. Please correct.

More substantive comments:

Line 157: Could the authors add text explaining why T_t and T_{sum} are not necessarily the same or perfectly correlated per Figures 4b and 4c? I think this would help readers avoid having to read the study cited in line 145 to better understand why the authors assert that the high pattern correlation validates their methodology in 158-159.

Line 243: In "Furthermore, an accurate representation of BMB aerosol types and interactions with clouds remain challenging due to the lack of observations, despite the tremendous advances in the past decades", I do not understand what the authors are referring to when they say "tremendous advances". Are the advances in modeling, observations, or something else? Please be explicit.

Lines 243-246: BMB aerosol-cloud interactions observations are limited but hardly lacking. I strongly suggest the authors acknowledge field campaign observations of BMB aerosol-clouds that have often included follow-up modeling efforts constrained by those observations. Below, I list several overview papers and a few more detailed studies as a starting point, but reviewing this literature and the citing/cited literature within those papers might help the authors write text that more accurately captures the state of the observational efforts in understanding the role of BMB aerosols in the climate system. Observational campaign overviews include (and likely are not limited to!):

1. SCAR-B <https://agupubs.onlinelibrary.wiley.com/doi/abs/10.1029/98JD02281> "Smoke, Clouds, and Radiation-Brazil (SCAR-B) experiment"
2. EXPRESSO <https://agupubs.onlinelibrary.wiley.com/doi/abs/10.1029/1999JD900291> "Experiment for Regional Sources and Sinks of Oxidants (EXPRESSO): An overview"
3. SAFARI2000 <https://agupubs.onlinelibrary.wiley.com/doi/full/10.1029/2003JD003747> "Africa burning: A thematic analysis of the Southern African Regional Science Initiative (SAFARI 2000)"
4. SAFARI2000 follow ups such as <https://rmets.onlinelibrary.wiley.com/doi/abs/10.1256/qj.03.100> "The effect of overlying absorbing aerosol layers on remote sensing retrievals of cloud effective radius and cloud optical depth" and <https://agupubs.onlinelibrary.wiley.com/doi/full/10.1029/2002JD002315> "Solar radiative forcing by biomass burning aerosol particles during SAFARI 2000: A case study based on measured aerosol and cloud properties"
5. AMMA <https://angeo.copernicus.org/articles/26/2569/2008/> "Large-scale overview of the summer monsoon over West Africa during the AMMA field experiment in 2006"
6. CLARIFY <https://acp.copernicus.org/articles/20/4059/2020/> "Open cells exhibit weaker entrainment of free-tropospheric biomass burning aerosol into the south-east Atlantic boundary layer"
7. ORACLES <https://acp.copernicus.org/articles/21/1507/2021/> "An overview of the ORACLES (ObseRvations of Aerosols above CLouds and their intERactionS) project: aerosol-cloud-radiation interactions in the southeast Atlantic basin"

Discussion: In the final statement of the discussion, the authors call for "worldwide observing networks and continuous efforts in laboratory experiments" to resolve issues. After reviewing the BMB related field campaigns I point to above, and then also acknowledging the role of existing observing networks such as NASA AERONET <https://aeronet.gsfc.nasa.gov/> and NOAA GML <https://gml.noaa.gov/ccgg/>, I would suggest that the authors revise this final statement so that their assertion does not imply the onus is not solely on the observational/lab community. The modeling of BMB and BMB-cloud interactions is, in my opinion, a very challenging problem in Earth

system science because we do not have a perfect grasp of the history (e.g. Van Marle et al 2017 and Lamarque et al 2010, as the authors cited, but perhaps also papers attempting to tie historical proxies with observations such as <https://bg.copernicus.org/articles/13/3225/2016/> "Reconstructions of biomass burning from sediment-charcoal records to improve data-model comparisons") or the physics, or the chemistry, or the transport and exchanges via atmospheric and carbon cycle dynamics. Note also that there was a dedicated modeling effort called FireMIP <https://gmd.copernicus.org/articles/10/1175/2017/> "The Fire Modeling Intercomparison Project (FireMIP), phase 1: experimental and analytical protocols with detailed model descriptions" that highlighted the challenges in fire modeling. My point is that the climate modeling community is forced to deal with all this uncertainty, so I would argue that there is a larger burden on the climate modeling community because this is where synthesis of the myriad of observations is required. I could concede that there is an equal burden between modeling and observational communities, but definitely find myself disagreeing with the authors final statement as it is written. My disagreement or the distribution of the burden are, of course, not as important as the fact that the authors did not acknowledge the BMB-cloud observations that are already published, which itself calls into question their final statement. For example, as a hypothetical, suppose we had perfect and complete observations of BMB globally and since the year 1850. Do the authors believe that even the most physically/chemically complex climate models could perfectly reproduce observations? In order to avoid distracting readers from the relatively focused findings that the authors presented regarding TIO and BMB, I would suggest the authors carefully revise their Discussion to more completely reflect the existing studies of BMB aerosols in the Earth system, and end on a note that highlights work required by both the observational/lab communities and large-scale modeling communities, and perhaps suggests that there is strength in collaboration among those communities.

Reviewer #3 (Remarks to the Author):

The paper titled "Critical role of biomass burning aerosols in enhanced historical Indian Ocean warming" by Tian et al uses the CESM individual forcing experiments to analyse the role of different radiative forcing changes to the global warming pattern. They find that changes in biomass burning contributes the most to a more rapidly warming Indian Ocean. The work shows interesting results and is generally well written, however there are a number of issues that should be cleared up before publication.

1. The words "significant" and "significantly" are use throughout the paper which implies the use of a significance test, yet this information is not provided (For example, Line 195 and the accompanying Fig. 5a). Information on the statistical significance of the results would greatly improve my confidence in this paper. As it stands, I'm not convinced the relative trends shown for BMB are even notable.
2. For the Impacts of BMB-induced TIO relative warming section I am not sure how the impact of the TIO is attributed to these results in addition to changes in BMB. For example, could the impact of BMB on the Atlantic Ocean impact the Atlantic jet-stream?
3. Line 198: There are precipitation changes, but a precipitation shift is not obvious to me in this plot, as the mean values are about 10 times larger than the century-scale trends. Again, the statistical significance of these trends is important.
4. Lines 106-109: I don't understand the standard deviation analysis, perhaps this can be solved with a better description? Are you finding the standard deviation of all the grid points in the ensemble mean trend field? Then wouldn't a lower standard deviation mean the trends are spatially more alike?
5. Lines 81-83: It is a difficult problem to know if ocean memory has been eliminated, perhaps you could say you eliminated sea surface memory?
6. The NAO index has not been defined.

Response to the reviews of the manuscript “Critical role of biomass burning aerosols in enhanced historical Indian Ocean warming”

By Yiqun Tian, Shineng Hu, Clara Deser

We thank the reviewers for their comments and suggestions that led to many improvements in the paper. In particular, we have conducted additional significance tests on our linear trend figures. The text was further edited for misprints and clarity. We feel that the revised draft is greatly improved thanks to the reviewers’ constructive comments. Our point-by-point responses to the reviews are below (reviewers’ comments in italic, authors’ responses in blue font).

Reviewer #1 (Remarks to the Author):

Present study investigates the possible reasons for the relative stronger surface warming over the tropical Indian Ocean than the tropical mean during the past century, and finds the important contribution from the changes of biomass burning (BMB) aerosols. The results are really interesting, and emphasizing the role of BMB is quite new and important to the climatology field. The whole paper is well organized, but I still have some concerns before the publication of this paper.

Major comments:

1. My major concern is the model, and only one model is used in present study. The model may have some problems in simulating the physical processes involved even if CESM1-LE can capture some of observational warming trend, which could be only a result of model bias compensation. Besides, as the authors mentioned, the aerosol’s direct and indirect effects are not well simulated, and aerosol is the major source of forcing uncertainty in IPCC report. Thus, whether the obtained results are model-dependent? Does there exist any other experiments or models that could be used to further repeat your results (i.e., PDRMIP)? To what extent the BMB contributes to the TIO relative warming could be credible?

We totally agree with the reviewer that the results presented in this study are based on one model, CESM1, and can be model dependent. Many climate models participating in CMIP have performed historical simulations isolating the aerosol forcing as a whole, but as far as we know, CESM seems to be the only model further isolating the BMB aerosols. If BMB simulations with other models become available in the future, we will analyze them for multi-model comparison. We should emphasize that CESM1 here is forced by aerosol forcing used for all CMIP5 models and that the simulated TIO relative warming is consistent with the BMB aerosol reduction over the TIO together with the BMB aerosol increase over the tropical Pacific and Atlantic. This strengthens our confidence that the BMB-induced TIO relative warming should be a robust result, although the sensitivity of such response could be model dependent. We have now added the following sentence highlighting the issue of model dependence at Lines 242-244:
“To what extent the results presented in this study are model dependent needs to be tested by other climate models conducting similar types of historical experiments isolating the contribution of BMB aerosol changes, which is currently not a protocol for most models.”

2. *Why is the role of BMB critical in the TIO relative warming? As shown in Fig. 3, the total aerosols' contribution is negative, even if the large positive contribution shown by the BMB. That is to say, the other aerosols, i.e., black carbon, should be the larger negative contribution than the positive contribution of BMB. Besides, why is the combined trend of GHG and aerosols far less than the original TIO relative warming trend? There seems to exist other reasons, which work and make up the large gap between the combined trend of GHG and aerosols and original TIO relative warming trend.*

Sorry about the confusion as we did not explain the experimental set-up clearly. The AER forcing in our previous manuscript, now renamed as “AAER” in the revised version, actually represents the aerosols only from anthropogenic sources, while the BMB represents the aerosols from biomass burning. The two do not have any overlap, and their addition represents the total effect of tropospheric aerosol forcing in CMIP5 models. Thus, the results of AAER and BMB are independent of each other. In addition, the types of BMB aerosols in the model include mainly black carbon, particulate organic matter, SO₂ and SO₄. More details on the experimental set-up can be found in Deser et al. (2020).

The combined effect of GHG and AAER on the TIO relative SST trend is far less than their individual effects because they largely cancel out (see Fig. 3d-f). GHG forcing alone leads to a relative warming in the northern TIO and a relative cooling in the southern TIO, while AAER forcing alone leads to a relative warming in the southern TIO and a relative cooling in the northern TIO. The GHG + AAER thus has much smaller relative SST trends in the TIO than GHG and AAER separately.

To avoid confusion, we have now renamed AER as AAER (anthropogenic aerosols) in the revised manuscript.

Minor comments:

L. 88-89: Remove “the relative warming of”.

It has been removed (Line 87).

L. 89-90: Why is the relative warming over the tropical western Pacific not well captured? Please explain and discuss.

It is a great question. We have noticed this model-observation mismatch in the western Pacific relative warming, among other mismatches such as in the southeastern tropical Pacific and Atlantic. One possible cause of those model-observation mismatches is that multidecadal internal variability could potentially contribute to the observed trend during 1925-2021 but is mostly absent in the ensemble-mean of CESM1-LE. Other possibilities include the uncertainties in radiative forcing (including greenhouse gases, anthropogenic and biomass burning aerosols, etc.) or the model's mean-state biases. We have now added a discussion on this at Lines 89-92.

L. 265: “greenhouse gases, (GHG)” should be “greenhouse gases (GHG),”?

Corrected (Line 281).

Reviewer #2 (Remarks to the Author):

I reviewed the manuscript with a focus on the attribution to the TIO relative warming due to changes in regional BMB aerosol emissions. The physical mechanisms for TIO warming due to regional changes in BMB aerosol emissions are well founded, and the authors acknowledge some of the ongoing areas of study related to BMB absorption. Studying the effect of changing BMB aerosol emissions on the temperature trends of TIO relative to the tropical oceans in general is a useful method. In general, I find the authors largely remained focused on exploring a singular question, with relatively straight-forward but interesting methods and findings, and solid visualizations/figures of the key elements contributing to the overall conclusions. My recommendation is that it should be published after minor revisions.

Simple comments:

Line 61 “rise of greenhouse gases” \diamond “increase in greenhouse gas concentration”; rise makes it sound like the greenhouse gases molecules are floating upwards.

The word “rise” could indeed cause ambiguity. This has now been revised as suggested (Line 60).

Line 136 “reduced” \diamond “decreased”

Revised (Line 137).

Lines 159-160: Would it be clearer to state that this analysis here is for BMB-induced TIO relative warming as opposed to just TIO relative warming?

Yes, this is very helpful in increasing clarity. Revised as suggested (Line 163).

Figure 4: In b-j, I suggest writing out the terms in the subtitles of the figures rather than just using the variable as the subtitle. For example, in Figure 4b, the subtitle would be “Simulated tropical SST trends induced by BMB changes (T_t)”. You have plenty of room to include this and that way the readers would largely be able to learn from the figure itself rather than having to connect the individual variables to the caption.

The subtitles have been added in Figure 4 as suggested.

References seem misaligned. Line 212 references ref24 when I think the authors mean ref25. Similarly, in the Discussion, the 80% of BB emissions statement in line 222 likely refers to Van Marle et al 2017 (ref26) but the authors cite ref25. Seems like everything is off by one reference number for some reason. Please correct.

Corrected. Thank you for catching these mismatches.

More substantive comments:

Line 157: Could the authors add text explaining why T_t and T_{sum} are not necessarily the same or perfectly correlated per Figures 4b and 4c? I think this would help readers avoid having to read the study cited in line 145 to better understand why the authors assert that the high pattern correlation validates their methodology in 158-159.

Great point. We have clarified this in the main text (Lines 151-153) and also in the Methods section (Lines 345-347) as follow.

“It is worth noting that the sum of the seven sub-terms (denoted as T_{sum}^t) may not be expected to exactly match the model simulated T^t due to the assumptions made through the derivation, e.g., a linearized bulk formula of latent heat flux.”

Line 243: In “Furthermore, an accurate representation of BMB aerosol types and interactions with clouds remain challenging due to the lack of observations, despite the tremendous advances in the past decades”, I do not understand what the authors are referring to when they say “tremendous advances”. Are the advances in modeling, observations, or something else? Please be explicit.

It was not stated clearly in our previous version. We meant the advances in both modeling and observations. We have now clarified this in the revised manuscript with more details, including references and discussions (Lines 247-249).

Lines 243-246: BMB aerosol-cloud interactions observations are limited but hardly lacking. I strongly suggest the authors acknowledge field campaign observations of BMB aerosol-clouds that have often included follow-up modeling efforts constrained by those observations. Below, I list several overview papers and a few more detailed studies as a starting point, but reviewing this literature and the citing/cited literature within those papers might help the authors write text that more accurately captures the state of the observational efforts in understanding the role of BMB aerosols in the climate system. Observational campaign overviews include (and likely are not limited to!):

1. SCAR-B <https://agupubs.onlinelibrary.wiley.com/doi/abs/10.1029/98JD02281> “Smoke, Clouds, and Radiation-Brazil (SCAR-B) experiment”
2. EXPRESSO <https://agupubs.onlinelibrary.wiley.com/doi/abs/10.1029/1999JD900291> “Experiment for Regional Sources and Sinks of Oxidants (EXPRESSO): An overview”
3. SAFARI2000 <https://agupubs.onlinelibrary.wiley.com/doi/full/10.1029/2003JD003747> “Africa burning: A thematic analysis of the Southern African Regional Science Initiative (SAFARI 2000)”
4. SAFARI2000 follow ups such as <https://rmets.onlinelibrary.wiley.com/doi/abs/10.1256/qj.03.100> “The effect of overlying absorbing aerosol layers on remote sensing retrievals of cloud effective radius and cloud optical depth” and <https://agupubs.onlinelibrary.wiley.com/doi/full/10.1029/2002JD002315> “Solar radiative forcing by biomass burning aerosol particles during SAFARI 2000: A case study based on measured aerosol and cloud properties”
5. AMMA <https://angeo.copernicus.org/articles/26/2569/2008/> “Large-scale overview of the summer monsoon over West Africa during the AMMA field experiment in 2006”

6. CLARIFY <https://acp.copernicus.org/articles/20/4059/2020/> “Open cells exhibit weaker entrainment of free-tropospheric biomass burning aerosol into the south-east Atlantic boundary layer”
7. ORACLES <https://acp.copernicus.org/articles/21/1507/2021/> “An overview of the ORACLES (ObseRvations of Aerosols above CLouds and their intEractionS) project: aerosol–cloud–radiation interactions in the southeast Atlantic basin”

Thank you for referring us to these important references on BMB field campaign observations and the associated modeling efforts. They are all very helpful and relevant to our study. We have now cited these references and added discussions on them in the main text (Lines 247-250).

Discussion: In the final statement of the discussion, the authors call for “worldwide observing networks and continuous efforts in laboratory experiments” to resolve issues. After reviewing the BMB related field campaigns I point to above, and then also acknowledging the role of existing observing networks such as NASA AERONET <https://aeronet.gsfc.nasa.gov/> and NOAA GML <https://gml.noaa.gov/ccgg/>, I would suggest that the authors revise this final statement so that their assertion does not imply the onus is not solely on the observational/lab community. The modeling of BMB and BMB-cloud interactions is, in my opinion, a very challenging problem in Earth system science because we do not have a perfect grasp of the history (e.g. Van Marle et al 2017 and Lamarque et al 2010, as the authors cited, but perhaps also papers attempting to tie historical proxies with observations such as <https://bg.copernicus.org/articles/13/3225/2016/> “Reconstructions of biomass burning from sediment-charcoal records to improve data–model comparisons”) or the physics, or the chemistry, or the transport and exchanges via atmospheric and carbon cycle dynamics. Note also that there was a dedicated modeling effort called FireMIP <https://gmd.copernicus.org/articles/10/1175/2017/> “The Fire Modeling Intercomparison Project (FireMIP), phase 1: experimental and analytical protocols with detailed model descriptions” that highlighted the challenges in fire modeling. My point is that the climate modeling community is forced to deal with all this uncertainty, so I would argue that there is a larger burden on the climate modeling community because this is where synthesis of the myriad of observations is required. I could concede that there is an equal burden between modeling and observational communities, but definitely find myself disagreeing with the authors final statement as it is written. My disagreement or the distribution of the burden are, of course, not as important as the fact that the authors did not acknowledge the BMB-cloud observations that are already published, which itself calls into question their final statement. For example, as a hypothetical, suppose we had perfect and complete observations of BMB globally and since the year 1850. Do the authors believe that even the most physically/chemically complex climate models could perfectly reproduce observations? In order to avoid distracting readers from the relatively focused findings that the authors presented regarding TIO and BMB, I would suggest the authors carefully revise their Discussion to more completely reflect the existing studies of BMB aerosols in the Earth system, and end on a note that highlights work required by both the observational/lab communities and large-scale modeling communities, and perhaps suggests that there is strength in collaboration among those communities.

Thank you for sharing your thoughts on the pressing challenges of simulating BMB aerosols in climate models. This thoughtful comment and the studies mentioned are all very helpful and

informative. We have now revised the Discussion and emphasized the challenge for modelers instead of only mentioning observational/lab communities (Lines 256-264) as follows.

“As continuous observational efforts remain necessary in the coming decades, developing proxies of historical BMB aerosols³⁸ and modeling their physical and chemical properties, transport within the atmosphere, interactions with clouds, and role in carbon cycle dynamics all need to be refined. In that regard, modeling efforts like the Fire Modeling Intercomparison Project³⁴ will be particularly helpful in interpreting and potentially reducing inter-model spread. Improving the understanding and simulation of BMB aerosols thus requires the integrated efforts of worldwide observational and laboratory communities, Earth science modeling communities, and more importantly, the collaboration between those communities.”

Reviewer #3 (Remarks to the Author):

The paper titled “Critical role of biomass burning aerosols in enhanced historical Indian Ocean warming” by Tian et al uses the CESM individual forcing experiments to analyse the role of different radiative forcing changes to the global warming pattern. They find that changes in biomass burning contributes the most to a more rapidly warming Indian Ocean. The work shows interesting results and is generally well written, however there are a number of issues that should be cleared up before publication.

1. The words “significant” and “significantly” are use throughout the paper which implies the use of a significance test, yet this information is not provided (For example, Line 195 and the accompanying Fig. 5a). Information on the statistical significance of the results would greatly improve my confidence in this paper. As it stands, I’m not convinced the relative trends shown for BMB are even notable.

Thank you for this comment, we have added significant tests with 99% confidence to all of the trend figures.

2. For the Impacts of BMB-induced TIO relative warming section I am not sure how the impact of the TIO is attributed to these results in addition to changes in BMB. For example, could the impact of BMB on the Atlantic Ocean impact the Atlantic jet-stream?

This is a great point. We should be careful when interpreting the BMB change-induced climate response. The BMB forcing in CESM1-LE is globally distributed. We can’t tell whether the strengthened jet stream (see 850 mb zonal wind trends in Fig. 5c) is impacted by the BMB effects on the TIO or also on the Atlantic Ocean. Previous studies suggest that a TIO relative warming can induce a positive phase of NAO through atmospheric teleconnections (Hoerling et al. 2001, 2004; Bader and Latif 2003; Hu and Fedorov 2020), and our study suggests that the historical TIO relative warming could at least partly be caused by the changes in BMB aerosols. Bridging the two links, we claimed the positive NAO-like response in the Atlantic jet-stream as the impact of BMB-induced TIO relative warming. We agree with the reviewer that we cannot rule out the possibility of other BMB-related factors contributing to the Atlantic jet-stream changes.

Therefore, we changed the section title to “Potential impacts of BMB-induced relative warming”. In addition, after discussing the mechanisms mentioned above, we added a sentence highlighting the possible contributions from other BMB-induced changes (Lines 221-224).

“A more definitive demonstration of the BMB-TIO-NAO linkage, in particular whether the BMB-induced Atlantic jet strengthening is explained solely by the TIO relative warming or partially by other factors, requires further investigation.”

3. Line 198: There are precipitation changes, but a precipitation shift is not obvious to me in this plot, as the mean values are about 10 times larger than the century-scale trends. Again, the statistical significance of these trends is important.

We agree with the reviewer that the precipitation changes are relatively small compared to the climatological precipitation rates. We conducted a significance test and found that only the areas that see the strongest rainfall change can pass the 99% significance test. This is due partly to the highly variable nature of precipitation and also to the relatively small ensemble size of XBMB simulations. Lowering the threshold of the significance test to 95% will increase the area passing the test, as expected (Fig. R1). But we decided to keep the 99% threshold to be consistent with the significance tests for the rest of paper. In the text, we deleted the word “significant” to be more conservative, and highlighted the fact that the rainfall change is rather weak (Lines 201-204).

Fig. R1: Fig. 5 with 95% confidence.

4. Lines 106-109: I don't understand the standard deviation analysis, perhaps this can be solved with a better description? Are you finding the standard deviation of all the grid points in the ensemble mean trend field? Then wouldn't a lower standard deviation mean the trends are spatially more alike?

We agree that our previous description was indeed confusing. We have now added explicitly the definition of spatial standard deviation and rephrased a few places to avoid confusion (Lines 106-111) as follows.

“Next, we compute the standard deviation of all the grid points in the ensemble-mean SST trend field and use it as a measure of spatial variance. Interestingly, we find that BMB contributes significantly to the spatial variability of tropical SST warming and is comparable to the contributions from GHG or AAER (Fig. 2b). As far as we know, the impact of BMB on the spatial variability of tropical SST trends (i.e., the pattern) has been overlooked in previous studies.”

5. *Lines 81-83: It is a difficult problem to know if ocean memory has been eliminated, perhaps you could say you eliminated sea surface memory?*

Thanks for your suggestion. We agree that the ocean memory, especially in the deep ocean, cannot be completely removed by choosing the starting year of 1925. Therefore, we rephrased the sentence as,

“...we use the period 1925-2005 for analysis to **remove some of** the potential influence of ocean initial condition memory.” (Lines 79-81)

6. *The NAO index has not been defined.*

We have now added the definition of the NAO index computed in our study in the Method section under Definitions of indices (Lines 301-303).

References

Bader, J., & Latif, M. The impact of decadal-scale Indian Ocean sea surface temperature anomalies on Sahelian rainfall and the North Atlantic Oscillation. *Geophysical Research Letters*, 30(22), 2169 (2003).

Hoerling, M. P., Hurrell, J. W., & Xu, T. Tropical origins for recent North Atlantic climate change. *Science* 292, 90-92 (2001).

Hoerling, M. P., Hurrell, J. W., Xu, T., Bates, G. T., & Phillips, A. S. Twentieth century North Atlantic climate change. Part II: Understanding the effect of Indian Ocean warming. *Clim. Dyn.* 23, 391-405 (2004).

Hu, S., & Fedorov, A. V. Indian Ocean warming as a driver of the North Atlantic warming hole. *Nat. Commun.* 11, 1-11 (2020).

REVIEWERS' COMMENTS

Reviewer #1 (Remarks to the Author):

The authors have addressed the previous comments well in the revision, and I do not have further comments.

Reviewer #2 (Remarks to the Author):

Thank you to the authors for working hard on their responses to my comments, and the comments from the other reviewers. I have no further suggestions or comments.

Reviewer #3 (Remarks to the Author):

The authors have adequately addressed the concerns raised in my previous comments. I therefore recommend publication of this manuscript.